# Ascorbic acid supports ex vivo generation of plasmacytoid dendritic cells from circulating hematopoietic stem cells

Anders Laustsen[1], Renée M van der Sluis[1,2], Albert Gris-Oliver[1], Sabina Sánchez Hernández[1], Ena Cemalovic[1,3,4], Hai Q Tang[5], Lars Henning Pedersen[1,5,6], Niels Uldbjerg[6], Martin R Jakobsen[1]\*[†], Rasmus O Bak[1,2]\*[†]

[1]Department of Biomedicine, Aarhus University, Aarhus, Denmark; [2]Aarhus Institute of Advanced Studies, Aarhus University, Aarhus, Denmark; [3]Centre of Molecular Inflammation Research, Department of Clinical and Molecular Medicine, Norwegian University of Science and Technology, Trondheim, Norway; [4]Clinic of Medicine, St. Olav's University Hospital, Trondheim, Norway; [5]Department of Obstetrics and Gynaecology, Aarhus University Hospital, Aarhus, Denmark; [6]Department of Clinical Medicine, Aarhus University, Aarhus, Denmark

**\*For correspondence:**
mrj@biomed.au.dk (MRJ);
bak@biomed.au.dk (ROB)

[†]These authors contributed equally to this work

**Abstract** Plasmacytoid dendritic cells (pDCs) constitute a rare type of immune cell with multifaceted functions, but their potential use as a cell-based immunotherapy is challenged by the scarce cell numbers that can be extracted from blood. Here, we systematically investigate culture parameters for generating pDCs from hematopoietic stem and progenitor cells (HSPCs). Using optimized conditions combined with implementation of HSPC pre-expansion, we generate an average of 465 million HSPC-derived pDCs (HSPC-pDCs) starting from 100,000 cord blood-derived HSPCs. Furthermore, we demonstrate that such protocol allows HSPC-pDC generation from whole-blood HSPCs, and these cells display a pDC phenotype and function. Using GMP-compliant medium, we observe a remarkable loss of TLR7/9 responses, which is rescued by ascorbic acid supplementation. Ascorbic acid induces transcriptional signatures associated with pDC-specific innate immune pathways, suggesting an undescribed role of ascorbic acid for pDC functionality. This constitutes the first protocol for generating pDCs from whole blood and lays the foundation for investigating HSPC-pDCs for cell-based immunotherapy.

## Introduction

Plasmacytoid dendritic cells (pDCs) represent a rare and unique type of immune cell that plays a central role particularly in the detection and control of viral infections. In addition to conventional dendritic cell (cDC) functions, pDCs are capable of producing high levels of type I interferon (IFN) upon exposure to virus-derived nucleic acids that are recognized by Toll-like receptor (TLR) 7 and TLR9 (*Swiecki and Colonna, 2015*). Though the signature cytokine secreted by activated pDCs are type I IFNs, pDCs also effectively produce other pro-inflammatory cytokines and chemokines such as IL-1β, IL-6, IL-8, TNFα, and ligands for CXCR3 (CXCL9, CXCL10, and CXCL11) (*Tel, 2012*; *van Beek, 2020*). Consequently, pDCs have emerged as key effectors and regulators within the immune system, and their implication within a number of diseases, as well as their potential clinical application, has become a topic of great interest. Several preclinical studies have confirmed the immunotherapeutic potential of pDCs for the treatment of cancer through a multifaceted stimulation of the immune system (*Aspord, 2014*; *Drobits, 2012*; *Wu, 2017*; *Belounis, 2020*). Importantly, two clinical trials

have shown that autologous tumor antigen-loaded pDCs induce antitumoral responses and significantly improved clinical outcome for melanoma and prostate cancer patients, respectively (*Tel, 2013*; *Westdorp, 2019*). In one of these trials, a mixture of pDCs and cDCs was used, and a follow-up comparison of these two cell types suggests that pDCs are superior to cDCs at attracting CD8+ T cells, γ/δ T cells, and CD56+ NK cells to sites of melanoma (*van Beek, 2020*). Overall, this indicates that pDC-based anticancer immunotherapy could be an alternative or supplement to current cancer immunotherapy. While attempts have been made to translate the use of pDCs into a clinical immunotherapeutic setting, their use has been severely impeded by their rarity within peripheral blood (0.1% ± 0.07% of PBMCs) (*Ueda, 2003*). Coupled with their short ex vivo survival, they are very difficult to study and modulate as a population (*Zhan, 2016*). To this end, a few efforts have been made to generate pDCs ex vivo by differentiation of CD34+ hematopoietic stem and progenitor cells (HSPCs) (*Diaz-Rodriguez, 2017*; *Thordardottir, 2017*). These studies demonstrate that HSPC-derived pDCs (HSPC-pDCs) can be generated from different sources of HSPCs, including cord blood (CB) and mobilized peripheral blood. Although some improvements in different methods of pDC generation have been achieved, adoptive transfer therapy of autologous HSPC-pDCs is still challenging due to low cell yield and the requirement for patients to undergo granulocyte colony-stimulating factor (G-CSF)-induced HSPC mobilization.

Recently, we reported a novel robust ex vivo setup for generating high numbers of pDCs from HSPCs (*Laustsen, 2018*). We identified that a combination of growth factors, cytokines, and small molecules (Flt3-L, TPO, SCF, IL-3, and SR1) supported HSPC expansion and differentiation into immature HSPC-pDCs. Following a 21- day culture period, an average of 35% of the culture was HSPC-pDCs, which could be enriched to near purity using immunomagnetic depletion of non-pDCs. Most importantly, we showed that to generate a mature and functional phenotype, the HSPC-pDCs culture required priming by exposure to type I and II IFNs (*Laustsen, 2018*).

To extend on our previous work, we here present a clinical-relevant strategy to increase HSPC expansion and promote the functionality and numbers of generated pDCs under current good manufacturing process (cGMP) standards. HSPC pre-expansion with the pyrimido-indole derivative UM171 combined with low-density (LD) culturing highly promoted expansion of HSPC-pDCs. We demonstrate that commercially available cGMP medium fails to produce HSPC-pDCs with functional capacity

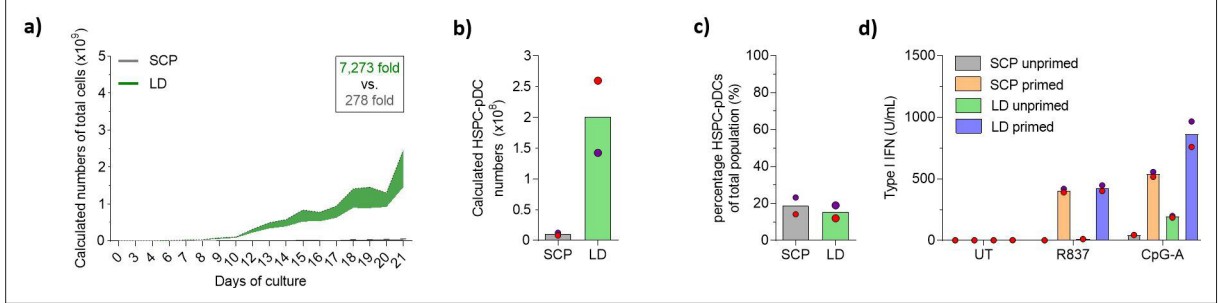

**Figure 1.** Lower cell density increases expansion during plasmacytoid dendritic cell (pDC) differentiation. Hematopoietic stem and progenitor cells (HSPCs) were thawed and 2 × 10⁵ cells were cultured for 21 days using the standard cultivation protocol (SCP) described previously, or with a low-density (LD) protocol (see *Figure 1—figure supplement 1a and b* for detailed culturing paradigms), after which HSPC-pDCs were isolated by immunomagnetic negative selection. (**a**) The total cumulative number of cells during culture was measured. To maintain culture format and minimize costs, a fraction of the culture was continuously discarded during passaging, which was taken into account when calculating cumulative cell numbers. (**b**) The number of HSPC-pDCs isolated after immunomagnetic negative selection was determined at day 21 and total cumulative number of HSPC-pDCs generated was calculated based on the fraction of cells discarded during culture. (**c**) Percentage of HSPC-pDCs of the total cell population at the day of isolation. (**d**) Isolated HSPC-pDCs were primed with type I IFN for 3 days or left unprimed, after which they were stimulated for 20 hr with agonists directed against TLR7 (R837) or TLR9 (CpG-A) and type I IFN was measured. Data shown represent mean of two cord blood donors.

The online version of this article includes the following source data and figure supplement(s) for figure 1:

**Source data 1.** Source data related to *Figure 1a-d*.

**Figure supplement 1.** Lower hematopoietic stem and progenitor cell (HSPC) density increases expansion of HSPCs during HSPC-derived plasmacytoid dendritic cell (HSPC-pDC) differentiation.

**Figure supplement 1—source data 1.** Source data related to *Figure 1—figure supplement 1g-h*.

to produce type I IFN upon TLR7 and TLR9 activation. Importantly, we identified that the addition of ascorbic acid (AA) upregulated several genes implicated in pDC development and function, and rescued the functionality of pDCs, thereby establishing AA as a key culture medium component for the generation of HSPC-pDCs. Finally, we show that these combined efforts enable the generation of pDCs from naturally circulating HSPCs found within peripheral blood (cHSPC). Collectively, we present a platform that allows cGMP-compliant generation of therapeutically relevant numbers of HSPC-pDCs from HSPCs obtained from standard blood samples without the need for mobilization regiments like G-CSF and plerixafor.

## Results

### Low-density HSPC culture improves expansion and yield of pDCs

Our previously published protocol for HSPC-to-pDC differentiation made use of a high-density culturing paradigm based on medium change on fixed days using fixed volumes of medium. Prior work has demonstrated that culturing HSPCs ex vivo at LD culture conditions stimulates the transition of HSPCs into the cell cycle, thereby supporting the expansion of HSPCs (*Charlesworth, 2018*). Therefore, we compared our standard culture protocol (SCP) to an LD protocol where cells were split more frequently and not allowed to reach a density exceeding $5 \times 10^6$ cells/mL at any time point (see *Figure 1—figure supplement 1a and b* for culturing overview). During the 21- day culture, we observed that the LD protocol led to a higher expansion of HSPCs compared to the standard protocol (*Figure 1a* and *Figure 1—figure supplement 1a–c*). Starting from $2 \times 10^5$ HSPCs, we obtained an average number of $1.5 \times 10^9$ ($\pm 0.7 \times 10^9$) total cells compared to $0.055 \times 10^9$ ($\pm 0.001 \times 10^9$) in the standard condition. Next, we isolated HSPC-pDCs using immunomagnetic depletion of non-pDCs. Importantly, we found that the LD culture protocol also highly increased HSPC-pDC numbers. Starting from $2 \times 10^5$ HSPCs, the average yield of isolated HSPC-pDCs was $201 \times 10^6$ ($\pm 58.7 \times 10^6$) versus $10.4 \times 10^6$ ($\pm 2.3 \times 10^6$) HSPC-pDCs in the standard condition (*Figure 1b*). This is an average yield of 1005 ($\pm 293$) HSPC-pDCs per single CD34$^+$ HSPC, which is an average improvement of 19-fold over the standard condition. The culture density did not affect the viability of the cells during the 21- day culture period (*Figure 1—figure supplement 1c*), nor did it influence the fraction of HSPC-pDCs among the total cells that were generated at day 21 (*Figure 1c*). However, a small decrease in viability of the isolated HSPC-pDCs from the LD culture was detected (*Figure 1—figure supplement 1d*). pDCs are known to secrete very large amounts of type I IFNs in response to agonists directed against TLR7 or TLR9 (*Swiecki and Colonna, 2015*). Previously, we have shown that isolated HSPC-pDCs require a priming by adding type I and II IFNs to the culture medium to become functionally active and responsive to TLR7 and TLR9 agonists. We therefore next primed the isolated HSPC-pDCs for 3 days and subsequently stimulated the cells with TLR7 and TLR9 agonists. Evaluation of type I IFN responses of HSPC-pDCs generated under LD conditions versus standard conditions showed no difference when stimulated with a TLR7 agonist and a minor improvement for the LD condition when stimulated with a TLR9 agonist (*Figure 1d*). Phenotypic analysis of pDC surface markers by flow cytometry of the isolated HSPC-pDCs (Lin$^-$CD11c$^-$CD123$^+$CD303$^+$) showed no obvious difference between the two culture conditions (*Figure 1—figure supplement 1e–h*).

### Serum-free medium improves expansion of HSPCs and increases pDC yield

Traditional cell culture medium remains a challenge for therapeutic use owing to the ill-defined and highly variable nature of serum. As we have previously shown that HSPC-pDCs could be generated using serum-free medium, we rationalized that combining commercially available serum-free medium (SFEM II medium) with our newly defined LD condition would constitute a more streamlined HSPC-pDC production protocol with possible improvements in functionality and yield (*Laustsen, 2018*). Additionally, we wanted to evaluate if the higher expansion rate of HSPCs in LD culture would enable earlier isolation of functional HSPC-pDCs, thereby reducing manufacturing cost and duration of a potential future clinical product. Of note, the high yield obtained with the LD protocol prompted us to reduce the starting cell numbers from $2 \times 10^5$ to $1 \times 10^5$ HSPCs and still generate sufficient numbers of HSPC-pDCs for subsequent analysis. We then compared SFEM II to the conventional serum-based culture medium (RPMI) at LD conditions (see *Figure 2—figure supplement 1a and b* for culture overview).

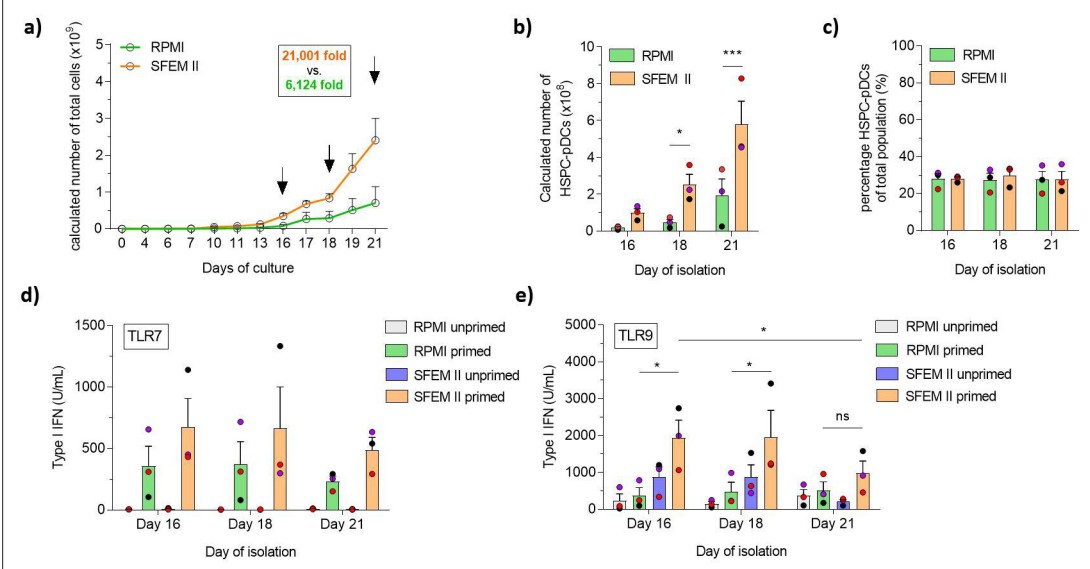

**Figure 2.** Serum-free conditions improve hematopoietic stem and progenitor cell-derived plasmacytoid dendritic cell (HSPC-pDC) yield, and cells isolated at earlier time points retain a functional phenotype. HSPCs were thawed and $1 \times 10^5$ cells were seeded in RPMI or SFEM II and cultured at a density of $0.5–5 \times 10^6$ cells/mL. HSPC-pDC isolation was performed after 16, 18, and 21 days of culture and HSPC-pDCs were phenotypically analyzed. (**a**) Calculated numbers of cells during culture. Arrows indicate days when HSPC-pDCs were isolated. (**b**) Numbers of isolated HSPC-pDCs. (**c**) Proportion HSPC-pDCs within the total population of cells at the day of isolation. (**d, e**) Type I interferon (IFN) responses of non-primed or primed HSPC-pDCs after activation with the TLR7 agonist R837 (**c**) or the TLR9 agonist CpG 2216 (**d**). Data shown represent ± SEM of three donors (**a-c**) and three donors each analyzed in technical triplicates (**d-e**).

The online version of this article includes the following source data and figure supplement(s) for figure 2:

**Source data 1.** Source data related to *Figure 2a-e*.

**Figure supplement 1.** Serum-free conditions improve expansion of hematopoietic stem and progenitor cells (HSPCs) and HSPC-derived plasmacytoid dendritic cells (HSPC-pDCs) isolated at earlier time points retain a functional phenotype.

**Figure supplement 1—source data 1.** Source data related to *Figure 2—figure supplement 1b-c and f-g*.

Cells were cultured for 21 days while HSPC-pDCs were isolated by immunomagnetic selection at days 16, 18, and 21. We found that SFEM II significantly promoted the proliferation of HSPCs compared to RPMI with serum (*Figure 2a*). Starting from $1 \times 10^5$ HSPCs, an average of $2.4 \times 10^9$ ($\pm 0.6 \times 10^9$) total cells were generated at day 21 with SFEM II versus $0.7 \times 10^9$ ($\pm 0.4 \times 10^9$) total cells for RPMI (*Figure 2a*). The high expansion rate using SFEM II also translated to a significantly higher yield of isolated HSPC-pDCs at all three time points of isolation, with a maximum of $580 \times 10^6$ HSPC-pDCs ($\pm 123 \times 10^6$) HSPC-pDCs isolated after 21 days of culture using SFEM II (*Figure 2b*). Even after 16 days of differentiation, $100 \times 10^6$ ($\pm 22.2 \times 10^6$) HSPC-pDCs could be isolated starting from only $0.1 \times 10^6$ HSPC. The transition to SFEM II did not affect the proportion of HSPC-pDCs in the cell culture as similar percentages of HSPC-pDCs were observed across all three isolation time points, indicating that pDCs differentiate continuously during the culture (*Figure 2c*). Importantly, no difference in viability of the isolated HSPC-pDCs was observed at any time point or condition (*Figure 2—figure supplement 1c*). We next performed TLR7 and TLR9 activation assays of the produced HSPC-pDCs, which showed that cells generated using SFEM II displayed a higher capacity to produce type I IFN (*Figure 2d and e*). Interestingly, we observed that HSPC-pDCs that had been isolated at earlier time points produced significantly more type I IFN upon TLR9 activation, suggesting that there is a balance between proliferation potential and the immunotherapeutic properties of HSPC-pDCs. Phenotypic analysis of the pDC-related surface markers CD123 and CD303 on lin⁻CD11c⁻ cells showed no differences across time points and conditions (*Figure 2—figure supplement 1d–g*). Taken together, our data show that HSPC expansion and functionality of isolated HSPC-pDC generation were improved using the serum-free medium SFEM II compared to RPMI supplemented with serum. This enables earlier isolation of HSPC-pDCs while improving functionality of the cells and preserving high cell yield.

## LD culture supplemented with the pyrimido-indole derivative UM171 allows HSPC pre-expansion

In the pursuit of increasing HSPC-pDC yield further, we next investigated HSPC pre-expansion before initiating the pDC differentiation protocol. A fundamental limitation to HSPC ex vivo culturing is the rapid differentiation of stem and progenitor cells, which in turn produces inhibitory feedback signals that limit stem cell self-renewal. Recent publications have found that ex vivo culturing of HSPCs at low densities combined with the small molecules UM171 and SR1 promotes self-renewal

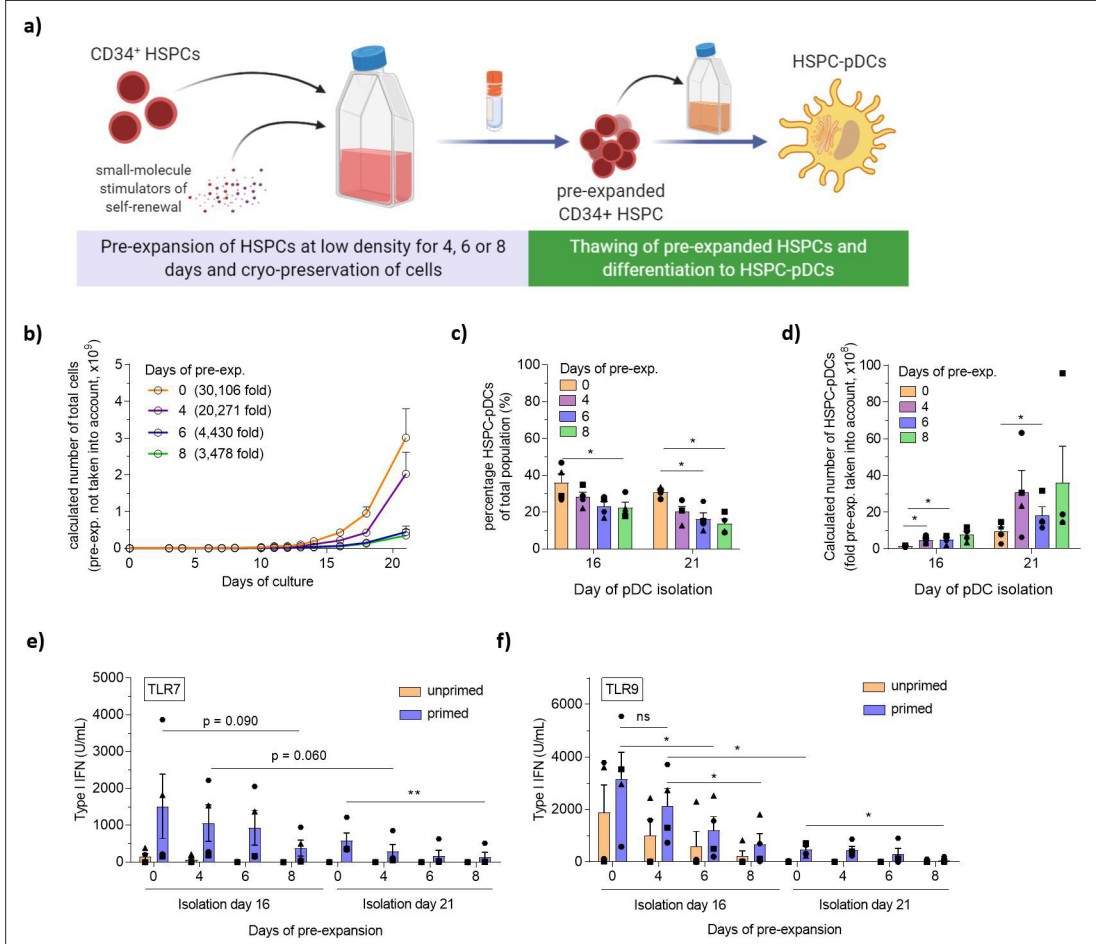

**Figure 3.** Pre-expansion of hematopoietic stem and progenitor cells (HSPCs) increases the yield of HSPC-derived plasmacytoid dendritic cells (HSPC-pDCs). (**a**) Schematic representation showing generation of HSPC-pDCs from pre-expanded HSPCs. HSPCs were pre-expanded at low density (1–5 × $10^5$ cells/mL) in SFEM II medium supplemented with UM171 for 4, 6, or 8 days and then cryopreserved. Cells were then thawed, phenotyped for CD34, and 1 × $10^5$ HSPCs were seeded for HSPC-pDC generation. HSPC-pDCs were isolated after either 16 or 21 days of culture and phenotypically analyzed. (**b**) Calculated number of cells during HSPC-pDC differentiation using pre-expanded HSPCs without the pre-expansion factor taken into account (same starting cell number at differentiation). (**c**) Percentage of HSPC-pDCs of total population of cells. (**d**) Calculated number of HSPC-pDCs isolated after 16 and 21 days of culture with fold pre-expansion taken into account. (**e, f**) Levels of type I interferon (IFN) from HSPC-pDCs after stimulation with the TLR7 agonist R837 (**e**) or the TLR9 agonist CpG-2216 (**f**). Data shown represent ± SEM of four donors (**a–d**) and four donors each analyzed in technical triplicates (**e, f**).

The online version of this article includes the following source data and figure supplement(s) for figure 3:

**Source data 1.** Source data related to *Figure 3b-f*.

**Figure supplement 1.** Pre-expansion of hematopoietic stem and progenitor cells (HSPCs).

**Figure supplement 1—source data 1.** Source data related to *Figure 3—figure supplement 1a-c and f-g*.

**Figure supplement 2.** Differention of pre-expanded hematopoietic stem and progenitor cells to HSPC-derived plasmacytoid dendritic cells (HSPC-pDCs).

**Figure supplement 2—source data 1.** Source data related to *Figure 3—figure supplement 2m-n*.

of primitive hematopoietic progenitor cells and long-term repopulating hematopoietic stem cells (LT-HSCs) (*Charlesworth, 2018*; *Fares, 2014*). Based on this, we set out to test if a pre-expansion HSPC protocol could be implemented prior to pDC differentiation, potentially allowing very limited numbers of CD34⁺ HSPCs to produce high yields of HSPC-pDCs (*Figure 3a*). HSPCs were cultured at LD concentrations (0.1–0.5 × 10⁶ cells/mL) for up to 8 days (*Figure 3—figure supplement 1a*), during which the cells expanded significantly with an average of 78 (± 14)-fold expansion for the 8- day pre-expansion (*Figure 3—figure supplement 1b, c*). Expanded HSPCs were cryopreserved after 4, 6, or 8 days of expansion to enable initiation of parallel pDC differentiation studies. Upon thawing, expanded HSPCs remained viable and positive (>95%) for the HSPC surface marker CD34⁺ with no difference compared to HSPCs that had not been expanded (*Figure 3—figure supplement 1d–h*). However, surface expression levels (MFI) of CD34 appeared to increase during the first four initial days of expansion, and then decrease again over time (*Figure 3—figure supplement 1h*). Next, using the established low-density SFEM II culture protocol, parallel pDC differentiations were initiated with equal numbers of HSPC-pDCs isolated after the different pre-expansion durations. We observed that HSPCs that had undergone pre-expansion for 6 and 8 days showed decreased expansion during pDC differentiation, but viabilities throughout differentiation remained high for all conditions (*Figure 3b* and *Figure 3—figure supplement 2i–k*). The stagnated growth correlated with increased numbers of adherent cells during later days of pDC differentiation, and these adherent cells displayed a morphology with long protrusions indicative of cDCs or macrophages (data not shown). This indicates that prolonged pre-expansion affects differentiation and the proliferative capacity of hematopoietic progenitors, which in turn may influence pDC differentiation. Accordingly, when initiating HSPC-pDC differentiation from the same starting cell numbers, lower yields of immunomagnetically separated HSPC-pDCs were observed upon prolonged pre-expansion, particularly when combined with 21 days of pDC differentiation (data not shown). A progressive decline in the percentage of HSPC-pDCs of the total cell population was also observed as the cells had been pre-expanded for longer durations (*Figure 3c*). Nevertheless, when fold pre-expansion was taken into account, very high numbers of pDCs could be generated with pre-expansion (*Figure 3d*). With a 4-day pre-expansion of HSPCs and a 21 -day pDCs differentiation, $3.1 \times 10^9$ ($\pm 1.2 \times 10^9$) HSPC-pDCs could be generated versus $0.9 \times 10^9$ ($\pm 0.3 \times 10^9$) HSPC-pDCs when no pre-expansion was applied. Similarly, early pDC isolation at day 16 yielded an average of $4.7 \times 10^8$ ($\pm 1.1 \times 10^8$) HSPC-pDCs when a 4- day pre-expansion was applied (*Figure 3d*). Next, we set up experiments where we primed the generated HSPC-pDCs and evaluated phenotypic and functional parameters of the cells. Interestingly, we found that the functional capacity of HSPC-pDCs to secrete type I IFN upon activation with TLR7 or TLR9 agonists was highly affected when cells were pre-expanded for a prolonged time (*Figure 3e and f*). This was in particular evident for the TLR9-mediated type I IFN response, which was significantly decreased when applying 6 or 8 days of pre-expansion. Conversely, surface expression of CD123 and CD303 was not affected by culture duration (*Figure 3—figure supplement 2l–n*). Overall, these data show a trend of reduced functionality with longer time in culture, which might reflect a type of functional exhaustion during prolonged cell culture. Collectively, we here show that prolonged pre-expansion, as well as extended pDC differentiation culture, drastically affects the functionality of generated HSPC-pDCs. Nevertheless, when a limited period of pre-expansion of 4 days is combined with an early isolation of HSPC-pDCs of 16 days, high yields of functional HSPC-pDCs can be generated.

## The use of a cGMP-compliant medium abolishes the ability of HSPC-pDCs to respond to TLR agonists

Recent technical advances in the use of defined media and synthetically made culture substrates have significantly simplified and improved the predictability of growing and differentiating stem cells. As clinical data highlight the promise of pDCs in immunotherapy (*Tel, 2013*), and since clinical cell products must be produced under cGMP, we pursued to implement a cGMP-compliant medium to our culture protocol. To that end, we performed pDC differentiation experiments using two commercially available cGMP-compliant serum-free media 'SCGM' and 'DC medium.' As we had already used a commercially available serum-free medium (SFEM II), we assumed an uncomplicated transition. Interestingly, we found that expansion of cells during pDC differentiation was highly reduced when HSPCs were cultured in cGMP medium (*Figure 4—figure supplement 1a*). However, cell viabilities remained high throughout the culture period, but the number of HSPC-pDCs that could be isolated from the

culture was reduced in accordance with the lower expansion (*Figure 4—figure supplement 1b and c*). Viabilities of isolated HSPC-pDCs and their proportion of the total cell population were largely unaffected (*Figure 4—figure supplement 1d and e*), leading us to believe that their functionality would remain unaffected. Remarkably, while isolated HSPC-pDCs from all three culture conditions phenotypically expressed the pDC markers CD123 and CD303 to the same extent, cells cultured in the two cGMP-compliant media were found to either completely lack or have drastically reduced capacity to produce type I IFN upon TLR7 or TLR9 agonist activation (*Figure 4—figure supplement 1f and g*).

## Ascorbic acid rescues the function of HSPC-pDCs produced using cGMP media

Vitamin C (ascorbic acid) is an essential vitamin in humans known to have pleiotropic functions in cellular biology, including immune cell function and hematopoiesis (*Manning, 2013*; *Huijskens, 2014*; *Huijskens, 2015*; *Bowie and O'Neill, 2000*; *Agathocleous, 2017*). Interestingly, AA has been shown to be involved in type I IFN immune responses. Unlike humans, mice can synthesize AA but transgenic mice lacking the capacity to synthesize AA actually show diminished capacity to produce type I IFN upon TLR activation and influenza infection. This indicates that AA may play a role in either TLR activity or pDC development (*Kim, 2013*; *Geber et al., 1975*). While no reports provide clear evidence of a role in pDC development of function, one study does find that medium supplement with AA can increase DC yield during ex vivo differentiation from HSPC, but its significance in pDC functionality was not investigated (*Thordardottir, 2017*). Given the evidence of AA in hematopoiesis and IFN responses, we hypothesized that adding AA to our culture medium would improve expansion and viability of HSPCs during pDC differentiation. To that end, we explored a culture system focusing on the cGMP-compliant 'DC medium' where we investigated the addition of physiological concentration of AA (50 µM) during HSPC-pDC differentiation (*Verrax and Calderon, 2009*). We found that DC medium supplemented with AA significantly promoted expansion during HSPC-pDC differentiation up to similar level as the SFEM II culture condition (expansion of 33,554-fold [± 11,664] for DC medium+ AA versus 13,456-fold [± 6953] for DC medium alone) (*Figure 4a* and *Figure 4—figure supplement 2h–i*). Moreover, addition of AA significantly improved viability of expanding HSPCs during culture, particularly during later stages of culture (day 12+) (*Figure 4—figure supplement 2j*). Additionally, we also found the viability of HSPC-pDCs isolated at day 21 to be significantly increased upon AA addition compared to DC medium alone (92.0% ± 3.4% versus 72.8% ± 3.2%) (*Figure 4—figure supplement 2k*). Accordingly, the yield of HSPC-pDCs was significantly increased upon AA supplementation for HSPC-pDCs isolated at day 21 ($1.0 \times 10^9 \pm 0.4 \times 10^9$ HSPC-pDCs for DC medium + AA versus $0.5 \times 10^9 \pm 0.3 \times 10^9$ for DC medium alone per $0.1 \times 10^6$ CD34$^+$ HSPCs), without affecting the percentage of HSPC-pDCs of the total population of cells (*Figure 4b and c*). No statistically significant differences were observed in the yield of HSPC-pDCs isolated at day 16 of culture for the different conditions. Next, we stimulated primed and unprimed HSPC-pDCs with synthetic TLR7 and TLR9 agonists and evaluated the type I IFN response. Interestingly, we found that HSPC-pDCs generated from the AA culture consistently elicited a robust type I IFN response in contrast to DC medium alone for cells isolated both on days 16 and 21 (*Figure 4d and e*). The response was even found to exceed the response of HSPC-pDCs generated using SFEM II medium. As previously observed, HSPC-pDCs isolated at 16 days of culture displayed improved capacity to produce type I IFN upon TLR7 or TLR9 activation compared to cells isolated at day 21 (*Figure 4d and e*). Moreover, we found that the capacity of HSPC-pDCs to secrete the pro-inflammatory proteins TNF-α and IL-6 upon TLR7 and TLR9 activation was also rescued when AA was present (*Figure 4—figure supplement 2l and m*). Upon analysis of surface expression of the markers CD123 and CD303, we found that pDCs isolated from AA-supplemented culture at day 16 displayed lower expression levels of CD303. However, no statistical significance was observed in the percentage of cells expressing CD303 between the conditions (*Figure 4—figure supplement 2n–p*). A small distinct population of HSPC-pDCs showing high surface expression of CD123 was observed upon AA supplementation (*Figure 4—figure supplement 2q*). This population was also observed for SFEM II culture conditions, albeit not to the same extent and not for all donors analyzed. Previously, we have shown that the pDC marker HLA-DR is highly expressed in HSPC-pDCs upon IFN priming, correlating with the capacity of the cells to activate T cells through antigen presentation (*Laustsen, 2018*). We similarly found

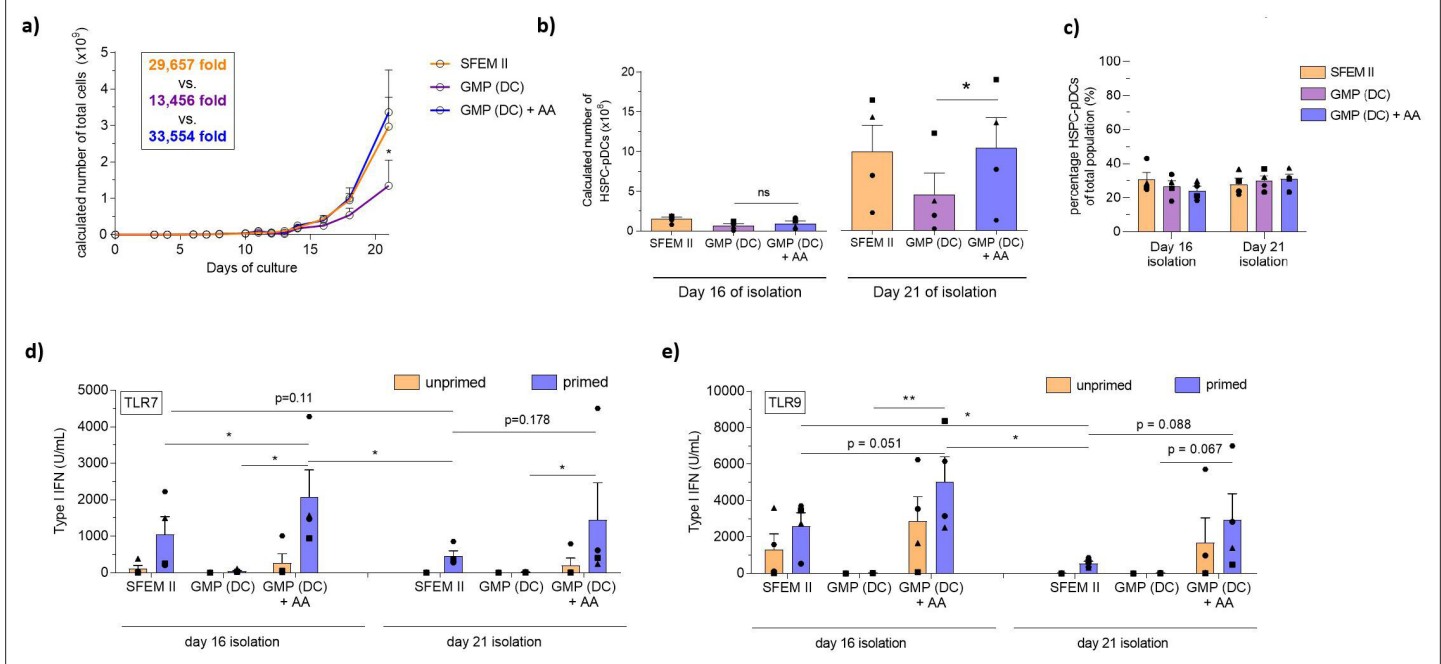

**Figure 4.** Ascorbic acid medium supplementation is required for hematopoietic stem and progenitor cell-derived plasmacytoid dendritic cell (HSPC-pDC) generation using the current good manufacturing process (cGMP)-compliant DC medium. HSPCs were thawed and $1 \times 10^5$ cells were seeded in SFEM II, the cGMP-compliant medium DC medium (GMP [DC]), or DC medium supplemented with ascorbic acid (GMP [DC] + AA). For all conditions, cells were kept at a density of 0.5–5 × $10^6$ cells/mL throughout culture. HSPC-pDCs were isolated after 16 and 21 days of culture and phenotypically analyzed. (**a**) Calculated number of total cells during HSPC-pDC differentiation. (**b**) Calculated number of HSPC-pDCs after isolation at 16 days or 21 days of culture. (**c**) Percentage of HSPC-pDCs of total cells. (**d, e**) Type I interferon (IFN) response of IFN primed or unprimed HSPC-pDCs isolated after 16 or 21 days of culture after activation with the TLR7 agonist R837 (**d**) or the TLR9 agonist CpG-2216 (**e**). Data shown represent ± SEM of four donors (**a–c**) and ± SEM of four donors each analyzed in technical triplicates (**d, e**).

The online version of this article includes the following source data and figure supplement(s) for figure 4:

**Source data 1.** Source data related to *Figure 4a-e*.

**Figure supplement 1.** cGMP compliant mediums fails to produce functional hematopoeitic stem and progenitor cell-derived plasmacytoid dendritic cells (HSPC-pDCs).

**Figure supplement 1—source data 1.** Source data related to Figure 4-figure supplement 1a-e and g.

**Figure supplement 2.** Ascorbic acid is required for generation of functional hematopoietic stem and progenitor cell-derived plasmacytoid dendritic cells (HSPC-pDCs) with DC medium.

**Figure supplement 2—source data 1.** Source data related to *Figure 4—figure supplement 2h, j-m,- o-p and s-t*.

HLA-DR to be upregulated upon IFN priming in HSPC-pDCs generated from DC medium. However, no statistically significant difference was observed upon AA addition (***Figure 4—figure supplement 2r and s***). Numerous studies have described that a distinct subset of precursor conventional dendritic cells (pre-DC) exists in peripheral blood (***See, 2017***). Due to functional and phenotypical similarities with blood pDCs, including surface expression of CD123, CD303, and CD304, and the capacity to produce type I IFN upon TLR7 or TLR9 agonist stimulation, this subpopulation has inadvertently been considered as pDCs (***Ito, 2006***; ***See, 2017***). Thus, we next evaluated if such pre-DCs were present in our HSPC-pDC population. Notably, pre-DCs have been shown to express CD2, CD5, and CD33, and secrete IL-12 upon TLR activation (***See, 2017***). However, HSPC-pDCs differentiated with or without AA were unable to produce IL-12 upon TLR7 and TLR9 agonist exposure and had no detectable levels of CD2 on the surface (***Figure 4—figure supplement 2t and u***). This supports that cDCs or pre-DCs are not present in the immunomagnetically selected HSPC-pDC population (***Laustsen, 2018***). Altogether, our results demonstrate that AA is essential for ex vivo differentiation of pDCs from HSPCs

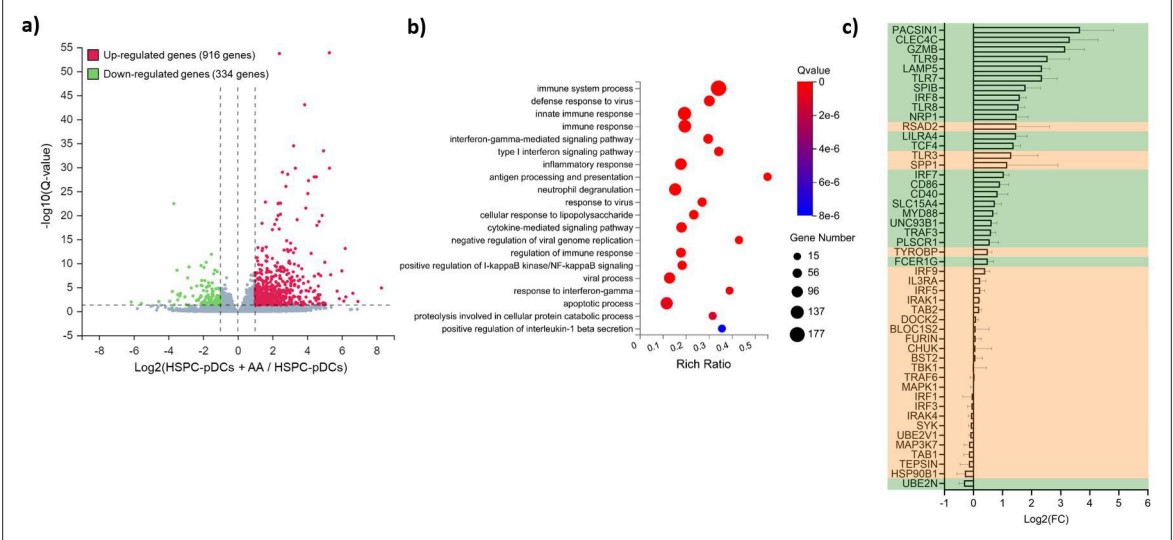

**Figure 5.** RNA-seq profile of hematopoietic stem and progenitor cell-derived plasmacytoid dendritic cells (HSPC-pDCs) generated with ascorbic acid (AA) shows upregulation of multiple genes related to pDC function. HSPCs were thawed and $1 \times 10^5$ cells were seeded in DC medium with or without supplementation of AA. Following 16 days of differentiation, pDCs were isolated by immunomagnetic depletion of non-pDCs, and HSPCs-pDCs were then primed for 72 hr with IFN, after which RNA was extracted and subjected to RNA-seq. (**a**) Volcano plot showing differentially expressed genes in the HSPC-pDCs generated with AA compared to HSPC-pDCs generated without AA. (**b**) Gene ontology bubble chart displaying the 20 most enriched biological processes for the differentially expressed genes in HSPC-pDCs generated with AA. The x-axis shows the enrichment ratio (rich ratio), which is the ratio between the number of differentially expressed genes within the biological process and the number of total genes annotated in that process. The size of the bubble represents the number of differentially expressed genes within the process, and the color represents the statistical significance of the enrichment. (**c**) Table showing differential expression of genes related to pDC development and function following addition of AA to the culture medium. The axis indicates log2 fold changes (log2FC) of gene expression in HSPC-pDCs generated with AA relative to cells generated without AA. Data shown are for three donors. Genes highlighted in green indicate significantly differentially expressed (Q-value below 0.05).

The online version of this article includes the following figure supplement(s) for figure 5:

**Figure supplement 1.** Ascorbic acid upregulates genes associated with antigen presentation.

when using cGMP-compliant medium by both increasing expansion of HSPCs and improving viability of expanding cells and isolated HSPC-pDCs, as well as functionality of isolated HSPC-pDCs.

## Ascorbic acid regulates numerous genes related to pDC development and immunological function

To examine the cellular effects of AA adjuvant to our HSPC-pDC culture system, we next performed transcriptome analysis of differentially expressed genes between IFN-primed HSPC-pDCs generated in DC medium with and without AA. Overall, we found that 1250 genes were differentially expressed upon AA addition; 916 genes upregulated and 334 genes downregulated (*Figure 5a*). A gene ontology analysis of the differentially expressed genes showed a significant enrichment of biological processes related to the immune system, including gene ontologies for innate immune sensing and responses to viruses (*Figure 5b*). Separately, we also found several genes associated with pDC development and function to be significantly upregulated by AA. In particular, we found the 'master regulator' of pDC development and function TCF4 (E2-2), and TCF-regulated genes, including SPIB, IRF7, IRF8, TLR7, TLR8, TLR9, UNC93B1, PACSIN-1, GMZB, CLEC4C, and LILRA4 to be significantly upregulated (*Figure 5c*; *Cisse, 2008*; *Cheng, 2015*; *Ghosh, 2010*). Moreover, we found several genes associated with innate immune functions of TLR7 and TLR9 to be activated upon AA addition. These included IRF5, IRF7, IRF8, TLR7, TLR9, MyD88, TRAF3, UNC93B1, LAMP5, PACSIN-1, RSAD2, SLC15A4, and SPP1 (*Figure 5c*). Of these genes, several are known to control assembly and trafficking of TLR7 and TLR9 to late endosomes, including Unc-93 homolog B1 (UNC93B1), phosholipase scrambase 1 (PLSCR1), solute carrier family 15 member A4 (SLC15A4), and BAD-LAMP (LAMP5) (*Combes, 2017*; *Blasius, 2010*; *Talukder, 2012*; *Pelka, 2018*). Of the remaining genes, myeloid differentiation primary response protein MyD88 (MyD88), protein kinase C and casein kinase substrate in neurons

1 (PACSIN1), interferon regulatory factor 5 (IRF5), IRF7, viperin (RSAD2), and osteopontin (SPP1) are directly involved in TLR7 and TLR9 downstream signaling, possibly explaining the observed lack of TLR7 or TLR9 activation upon absence of AA (*Figure 5c*; *Saitoh, 2011*; *Kawai, 2004*; *Honda, 2005*; *Shinohara, 2006*; *Schoenemeyer, 2005*). General surface markers specifically associated with pDCs were similarly found to be upregulated by AA, including CD303 (CLEC4C), CD304 (NRP1), CD123 (IL3RA), and CD85g (LILRA4) (*Figure 5c*). We also found multiple genes associated with antigen processing and presentation to be elevated transcriptional upon AA addition, particularly CD40, CD86, and genes within the HLA family of receptors (*Figure 5c* and *Figure 5—figure supplement 1a*). Since AA is a strong antioxidant and scavenges reactive oxygen species (ROS), we finally analyzed gene ontologies relating to ROS and oxidative stress. However, we did not identify an enrichment for any related differentially expressed genes (*Figure 5—figure supplement 1b*). Collectively, these data show that AA upregulates multiple genes associated with pDC development and immunological functions.

## Generation of HSPC-pDCs from whole blood

To this date, two clinical trials have reported the use of autologous peripheral blood-derived pDCs as a cell-based cancer immunotherapy. In both trials, pDCs were found to induce favorable anti-tumoral responses by effectively promoting anti-tumoral responses while being well-tolerated. Despite the use of leukapheresis to extract pDCs from the blood, the low numbers of peripheral blood pDCs that can be isolated remains a key limiting factor. In contrast, high numbers of pDCs can be generated by differentiation of patient-derived HSPCs isolated either by direct bone marrow aspiration or by blood leukapheresis following administration of mobilizing regiments (*Croop, 2000*). However, these procedures are invasive, painful, associated with side effects, or require multi-day doses of mobilizing drugs. For research purposes, for example, to investigate HSPC-pDC functions and antigen presentation to autologous memory cells, HSPCs acquired by these procedures are costly and can be a challenge to procure. An alternative source for HSPCs is peripheral whole blood where a limited number of naturally circulating CD34$^+$ HSPCs (cHSPCs) can be found. However, the rarity of these cells has so far limited their use for therapeutic purposes, and furthermore, their capacity for self-renewal and differentiation capacity has also been reported to be much lower than other sources of HSPCs (*De Bruyn, 2000*; *da Silva, 2009*). Based on our data, we reasoned that our high-yield differentiation protocol would allow therapeutically relevant numbers of HSPC-pDCs to be generated from cHSPCs. To that end, we obtained buffy coats (from around 450 mL of whole blood) and isolated CD34$^+$ cHSPCs using CD34 immunomagnetic-positive selection. We obtained an average number of $1.1 \times 10^6$ CD34$^+$ cHSPCs ( $\pm 0.6 \times 10^6$ cHSPCs), corresponding to 2457 cHSPCs (± 1307 cHSPCs) per mL of blood in line with previous observations (*Beyer and Muench, 2017*; *Eidenschink, 2012*; *Figure 6—figure supplement 1a*). Next, we systematically evaluated if a pre-expansion of cHSPCs and subsequent pDC generation would be feasible. We therefore initiated parallel 16- day pDC differentiation cultures of cryopreserved cHSPCs from the same donors that had either not been pre-expanded or pre-expanded for 4 days and subsequently cryopreserved. For a fully cGMP-compliant protocol, pre-expansion was performed using cGMP-compliant SCGM medium and pDC differentiation was performed using cGMP-compliant DC medium + AA. During pre-expansion, cHSPCs expanded 4.6-fold (± 1.5 -fold) and retained CD34 expression (*Figure 6—figure supplement 1b–j*). During the 16 days of pDC differentiation, we observed a 257-fold (± 91 -fold) expansion of total cells for non-pre-expanded cHSPC versus 192-fold (± 126 -fold) for cHSPC pre-expanded for 4 days (*Figure 6a*). When starting from $1 \times 10^5$ cHSPCs, the total yield of cHSPC-pDCs following immunomagnetic selection was $3.5 \times 10^6$ cHSPC-pDCs (± $2.9 \times 10^6$) without pre-expansion. Using an optimized setup with 4 days of pre-expansion, an average yield of $8.0 \times 10^6$ cHSPC-pDCs (± $4.5 \times 10^6$) was observed (*Figure 6b*). For the pre-expanded condition, this corresponds to 80 cHSPC-pDCs generated per single cHSPC. Of the total cell population generated, cHSPC-pDCs accounted for an average of 23% for pre-expanded cHSPC (*Figure 6c*). As expected, the observed pre-expansion, differentiation potential, and yield were less for cHSPCs compared to CB-HSPCs, whereas the frequency of pDCs obtained was similar. Importantly, cHSPC-pDCs were capable of producing type I IFN upon TLR7 or TLR9 stimulation, and the cells displayed a pDC surface phenotype similar to HSPC-pDCs derived from CB (*Figure 6d and e* and *Figure 6—figure supplement 1k–n*). As we had previously observed, AA was required for cHSPC-pDC TLR7 and TLR9-mediated type I IFN production (*Figure 6—figure*

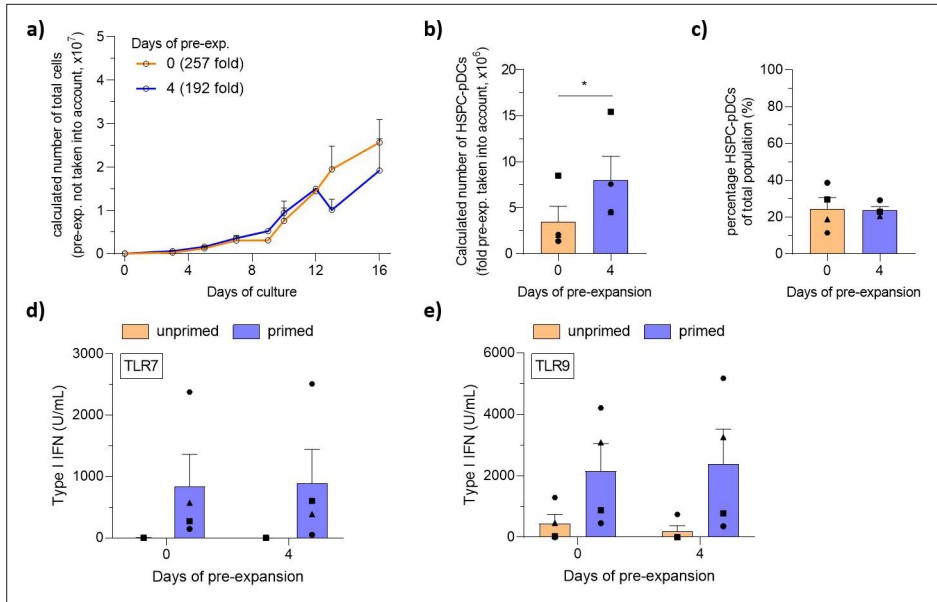

**Figure 6.** Generation of hematopoietic stem and progenitor cell-derived plasmacytoid dendritic cells from circulating HSPCs (cHSPCs) from peripheral whole blood using optimized current good manufacturing process (cGMP)-compliant medium. cHSPCs were pre-expanded for 4 days at low density ($1–5 \times 10^5$ cells/mL) in cGMP-compliant medium (SCGM) supplemented with UM171 and then cryopreserved. Subsequently, cells were thawed, phenotyped for CD34, and $1 \times 10^5$ cHSPCs were seeded for HSPC-pDC generation. HSPC-pDCs were isolated after 16 days of culture and phenotypically analyzed. (**a**) Calculated number of cells during HSPC-pDC differentiation using pre-expanded HSPCs (without the pre-expansion factor taken into account). (**b**) Calculated number of HSPC-pDCs upon isolation of HSPC-pDCs at 16 days of culture (with fold pre-expansion taken into account). (**c**) Percentage of HSPC-pDCs of the total population of cells. (**d, e**) Levels of type I IFN upon stimulation of HSPC-pDCs with the TLR7 agonist R837 (**d**) or the TLR9 agonist CpG-2216 (**e**). Data shown represent ± SEM of four donors (**a–c**) and four donors each analyzed in technical triplicates (**d, e**).

The online version of this article includes the following source data and figure supplement(s) for figure 6:

**Source data 1.** Source data related to *Figure 6a-e*.

**Figure supplement 1.** Differentiation of circulating CD34[+] hematopoietic stem and progenitor cell-derived plasmacytoid dendritic cells (cHSPC-pDCs) from cHSPCs using optimized current good manufacturing process (cGMP)-compliant conditions.

**Figure supplement 1—source data 1.** Source data related to *Figure 6—figure supplement 1a-d, f-j, m-o*.

---

*supplement 1o*). Collectively, we demonstrate that high numbers of functional HSPC-pDCs can be generated from a simple blood sample, which highly simplifies the procedure for generating pDCs for basic studies of pDC biology and for immunotherapeutic purposes.

## Discussion

Here we developed a new, robust, and simplified procedure for generating therapeutically relevant numbers of HSPC-pDCs from very limited numbers of HSPCs using cGMP-compliant medium. We found that differentiating HSPC-pDCs at reduced density and the supplementation of AA to the cGMP medium was key for achieving high and consistent numbers of highly functional HSPC-pDCs. This was underscored by the induction of several pDC signature genes by AA on the transcriptional level. Importantly, we showed that HSPC-pDCs could be generated ex vivo using HSPCs from whole blood. Collectively, our findings lay the foundation for further clinical exploration of pDCs for use as a cellular immunotherapy.

In the last decade, immunotherapy has emerged as a powerful strategy to treat multiple diseases. As the field has attracted a considerable interest from the pharmaceutical industry, the demand for

developing new methods and strategies is increasing. pDCs have received much attention owing to their multifaceted role in the immune system, and therapies that selectively activate pDCs, for example, TLR agonist, have proven to be effective in anti-tumoral therapy (*Iwahashi, 2010*; *Hofmann, 2008*; *Dummer, 2008*). However, while the importance of studying and modulating pDCs for therapy has become more evident, the progress has also been hampered by the low number of cells that can be extracted from the blood. This has also limited the use of pDCs for immunotherapy, but importantly, two independent clinical trials using adoptive transfer of autologous pDCs have shown clinical benefit (*Tel, 2013*; *Westdorp, 2019*). In one phase I clinical trial, Tel et al. vaccinated stage IV melanoma patients with autologous pDCs loaded with tumor peptides derived from the melanoma-associated antigens, gp100 and tyrosinase. The therapy improved overall survival, with 45% of patients still being alive after 2 years, compared to 10% in the matched control patients treated with conventional chemotherapy (*Tel, 2013*). Similarly, Westdorp et al. found in a phase IIa clinical trial that vaccination using cDCs and pDCs in combination, loaded with the tumor-associated antigens NY-ESO-1, MAGE-C2, and MUC1, improved the clinical outcome of patients with castration-resistant prostate cancer (*Westdorp, 2019*). In both clinical trials, antigen-loaded pDCs were found to effectively induce B and CD8$^+$ T cell anti-tumor-specific responses in patients, while being well-tolerated and safe (grade 1–2 toxicities) (*Tel, 2013*; *Westdorp, 2019*).

The principal drawback, which was also highlighted in these studies, was the maximum feasible dose of only 0.3–3 × 10$^6$ pDCs per vaccination, and only three vaccinations at biweekly intervals were performed (*Tel, 2013*; *Westdorp, 2019*). In contrast, in clinical trials utilizing monocyte-derived cDCs (moDCs), patients received 4–8 vaccine regiments with biweekly intervals with predefined moDC doses ranging from 10 to 30 × 10$^6$ cells (*Boudewijns, 2020*; *Okada, 2011*; *Ribas, 2010*). Similarly, the FDA-approved autologous dendritic cell immunotherapy, Provenge (sipuleucel-T), comprises three vaccination doses of a minimum of 50 million dendritic cells each, and some patients have been treated with doses as high as 1.3 billion dendritic cells (*Small, 2000*). Consequently, there is an unmet need for a clinical manufacturing protocol that allows high and robust numbers of pDCs to be generated. We believe that our platform meets this need by allowing the generation of consistent high numbers of autologous HSPC-pDCs that can be used for multiple vaccine regiments and that can be extracted from easily accessible whole blood. We and other groups have previously demonstrated that pDCs can be generated using CB CD34$^+$ HSPCs or mobilized peripheral blood CD34$^+$ HSPCs (mPB-HSPCs; *Diaz-Rodriguez, 2017*; *Thordardottir, 2017*; *Demoulin, 2012*; *Olivier, 2006*; *Curti, 2001*; *Thordardottir, 2014*). While CB HSPCs possess a great stem cell potential, the

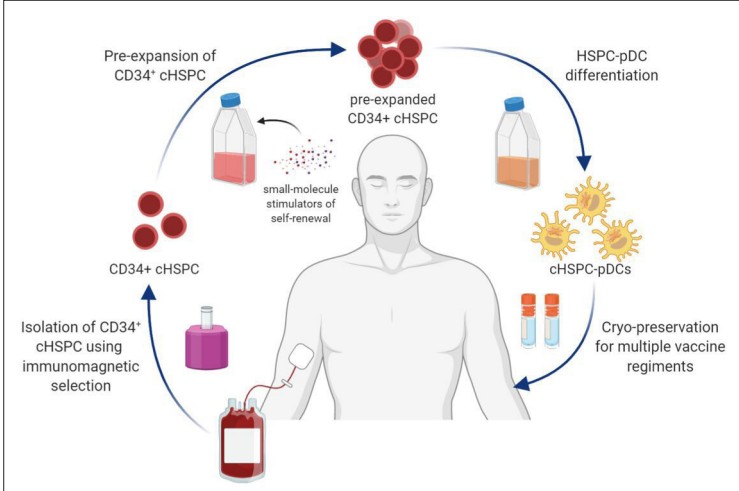

**Figure 7.** Schematic illustration showing the collective procedure of generating circulating CD34$^+$ hematopoietic stem and progenitor cell-derived plasmacytoid dendritic cells (cHSPC-pDCs) for therapeutic purposes starting from a patient blood sample. CD34$^+$ cHSPCs are initially isolated using immunomagnetic selection. cHSPCs are then pre-expanded at low density using small molecules that promote self-renewal. Subsequently, pre-expanded cHSPCs are differentiated into cHSPC-pDCs, which can either be readily used for immunotherapeutic purposes or cryopreserved to allow for multiple vaccine regiments.

major drawback is the need for an HLA match between donor and recipient. Obtaining mPB-HSPCs requires multiple injections of G-CSF usually over four consecutive days, followed by apheresis and large-scale CD34 immunomagnetic selection. Thus, the procedure is time-consuming, costly, requires access to expensive equipment, and is associated with inconvenience to the donor and side effects such as bone pain. For research studies of pDC biology, the same challenges apply. Furthermore, CB or mobilized peripheral blood is not easily available for common research laboratories. Natural pDCs from peripheral blood have been used, but very few numbers can be isolated from a buffy coat, and the cells have a half-life of only about 24 hr when cultured ex vivo (*Grouard, 1997*). We have previously shown that generated HSPC-pDCs show superior survival versus peripheral blood-derived pDCs, indicating that HSPC-pDCs are more suitable to receive modifications, for example, antigen loading and activation, which are required for immunotherapy in a clinical setting. Thordardottir et al. previously reported a yield of 1.6 million HSPC-pDCs starting from 100,000 mPB-HSPCs (*Thordardottir, 2017*). Using the same starting cell number and same duration of pDC differentiation, with no pre-expansion included, we generated an average of 306 million HSPC-pDCs amounting to a 191-fold improvement, albeit their starting material was mPB-HSPCs and ours CB-HSPCs. Importantly, we found that isolating pDCs earlier improved their capacity to produce type I IFN upon stimulation with synthetic TLR7 or TLR9 agonists. This indicates that prolonged pDC differentiation leads to a dysfunctional type I IFN response, possibly due to pDC exhaustion, albeit more research needs to be conducted to further characterize this phenotype. Using our shortened 16- day differentiation protocol and no HSPC pre-expansion, we generated up to 152 million HSPC-pDCs from 100,000 CB-HSPCs.

We believe that our findings can be used for cGMP-compliant manufacturing of clinically relevant numbers of autologous pDCs from a standard unit of whole blood (*Figure 7*). Using our pre-expansion strategy, we were able to generate an average of 8 million HSPC-pDCs starting from 100,000 cHSPCs. With an average number of $1.1 \times 10^6$ cHSPCs in a standard buffy coat from 450 mL of blood, this would allow for an average of 88 million HSPC-pDCs to be generated in a manner that is minimally invasive to the patient. The cHSPC-pDCs can in turn be cryopreserved, allowing repeated vaccine regiments (*Figure 7*). These numbers of generated cells highly exceed the number of natural pDCs that can be obtained from peripheral blood even when using leukapheresis (*Tel, 2013*; *Westdorp, 2019*). An additional advantage of HSPC-pDCs over natural pDCs is that HSPC-pDCs are amenable to genetic modifications (*Laustsen, 2018*). This potentially allows CRISPR/Cas gene editing to amplify the response of pDCs or render them resistant to inhibitory tumor signals. However, further preclinical studies are warranted on the potency of the generated HSPC-pDCs to induce anti-tumoral responses before the protocol can be translated and implemented into clinical practice.

In experiments with cGMP medium, we found that AA highly promoted viability of isolated HSPC-pDCs, and more importantly, that AA supplementation was crucial for the type I IFN response to TLR7 and TLR9 agonists. One study has previously reported that AA promoted the yield of pDCs differentiated from HSPCs (*Thordardottir, 2017*), but the effect of AA on pDC function was not assessed. Our observations indicate a hitherto undescribed key role of AA in pDC biology. Of note, AA is not included in conventional RPMI, but is present within serum, possibly explaining why pDCs generated using RPMI supplemented with fetal calf serum (FCS) displayed functional type I IFN responses. AA is highly unstable with sensitivity to temperature, atmospheric oxygen, light, and pH, which may explain some of the observed variations in the type I IFN responses (*Vojdani, 2000*). Some evidence indicates that AA plays a role in TLR function and innate immune function (*Bowie and O'Neill, 2000*; *Kim, 2013*; *Savini, 2002*; *Carter and Kane, 2004*; *Janovec, 2018*; *Esashi, 2012*). Gulo-/- mice, which cannot synthesize AA, show increased viral titers in the lung after infection with H3N2 influenza virus. Interestingly, the type I IFN response was found to be severely decreased, while pro-inflammatory cytokines, such as TNF-α and IL-1α/β, were increased, suggesting that AA plays a role in TLR function or downstream IFN signaling (*Kim, 2013*). Accordingly, Gulo-/- mice have been shown to have decreased STAT3 phosphorylation in T cells upon IL-22 stimulation (*Bae, 2013*). Moreover, AA has been shown to regulate the activity of the protein kinase C (PKC) family of serine/threonine kinases, of which some, including mitogen-activated protein kinase (MAPK), regulate TLR7 and TLR9-mediated type I IFN production (*Savini, 2002*; *Carter and Kane, 2004*; *Janovec, 2018*; *Esashi, 2012*). In line with the effect of AA on the TLR pathway, our transcriptome analysis revealed that several genes implicated in TLR7 and TLR9 processing, trafficking, and signaling were significantly upregulated by

AA. Thus, the observed transcriptional changes in these innate immune pathways possibly explain the rescue of type I IFN and pro-inflammatory responses we observed upon AA addition.

AA has been reported to play important roles in hematopoiesis and immune cell production and function (*Manning, 2013*; *Huijskens, 2014*; *Huijskens, 2015*; *Bowie and O'Neill, 2000*; *Agathocleous, 2017*; *Vojdani, 2000*; *Carr and Maggini, 2017*). Moreover, analyses of HSPCs isolated directly from the bone marrow has shown that they contain very high intracellular levels of AA (14-fold higher than physiological concentrations; *Agathocleous, 2017*). While AA supplementation increased HSPC-pDC yield in our setup, the overall percentage of HSPC-pDCs of the total cell population did not increase, indicating that AA does not specifically promote pDC differentiation. Nonetheless, we found several pDC signature genes, and genes related to pDC development, to be upregulated by AA. Especially we found TCF4 (E2-2), the key transcriptional regulator of pDC development and function, as well as other downstream E2-2-regulated genes quite relevant (*Cisse, 2008*; *Cheng, 2015*; *Ghosh, 2010*). The importance of E2-2 for pDC functionality has been clearly demonstrated in patients with Pitt–Hopkins syndrome. This disease is due to a monoallelic loss of E2-2, supporting defective pDCs with decreased CD303 (CLEC4C), and CD85g (LILRA4), and diminished capacity to produce type I IFN upon TLR9 activation (*Cisse, 2008*). Of other E2-2-regulated genes, SPIB and IRF8 have been shown to control downstream pDC commitment, supporting that AA is a key contributor in pDC development and function (*Sasaki, 2012*; *Schotte, 2004*; *Nagasawa, 2008*).

AA is a potent antioxidant that quenches harmful ROS and serves as a co-factor for many enzymes. Recent studies have shown that growth factors, such as TPO and IL-3, induce the formation of ROS in hematopoietic cells (*Sattler, 1999*; *Hensley, 2000*; *Carcamo et al., 2002*). ROS are well-known regulators of many biological pathways in cells, including cell differentiation (*Sattler, 1999*; *Hensley, 2000*; *Carcamo et al., 2002*). As TPO and IL-3 are included in our culture conditions, it is possible that ROS builds up during the prolonged culture period of the cells, thus affecting our pDC differentiation by prolonged culture. However, we did not observe any significant effects of AA on genes related to ROS and oxidative stress, suggesting that the phenotype observed is driven mainly by AA's regulation of pDC-related transcription factors and pro-inflammatory response-related genes. Further research into the exact mechanisms behind AA effects on this phenomenon needs to be conducted.

Collectively, we here demonstrate a clinically applicable stem cell differentiation procedure, which we believe can both help elucidate unresolved aspects of pDC biology and facilitate translation of a novel pDC-based treatment modality into clinical immunotherapy.

## Materials and methods
### Generation of HSPC-pDCs from CD34$^+$ HSPCs

Depending on optimization regiments, freshly or cryopreserved HSPCs from various sources were differentiated to HSPC-pDCs using different types of medium. For STD and LD/RPMI conditions, HSPCs were cultured in RPMI 1640 (Lonza) supplemented with 10% heat-inactivated FCS (HyClone), 600 µg/mL L-glutamine (Sigma), 200 U/mL penicillin, and 100 µg/mL streptomycin (Gibco, Life Technologies). For STD conditions, HSPCs were cultured at a fixed volume during pDC differentiation versus a fixed density of 0.5–5 × 10$^6$ cells/mL for the LD condition. For serum-free (SFEM II) or cGMP serum-free medium (DC medium) conditions, HSPCs were cultured in SFEM II or DC medium at LD (0.5–5 × 10$^6$ cells/mL). For all conditions, medium was supplemented with the cytokines and growth factors Flt3-L (100 ng/mL), SCG (100 ng/mL), TPO (50 ng/mL), and IL-3 (20 ng/mL). In contrast to our previous protocol, we omitted the addition of IL-7 as we showed that this did not positively impact HSPC-pDC generation (PMID 30166549). All cytokines are from PeproTech. In addition, the small molecule inhibitor StemRegenin 1 (SR1, STEMCELL Technologies) was added at a concentration of 1 µM. For serum-free medium conditions, a concentration of 20 µg/mL streptomycin and 20 U/mL penicillin (Gibco, Life Technologies) was added. Moreover, for the cGMP DC medium condition, 50 µg/mL of AA was added. Cells were cultured at 37 °C, 95% humidity, and 5% CO$_2$ for up to 21 days depending on optimizations. For the fixed density conditions, medium was replenished every 2–4 days depending on the growth of the HSPCs. Total cell numbers during expansion and differentiation of HSPCs were determined using a TC20 Automated Cell Counter (Bio-Rad). At the end of culture, HSPC-pDCs were enriched using a negative selection kit, according to the manufacturer's instructions (EasySep Human Plasmacytoid Dendritic Cell Enrichment kit, STEMCELL Technologies).

## Isolation of HSPC from CB

Deidentified umbilical cord blood (UCB) samples were obtained following scheduled cesarean section deliveries of healthy infants at the Department of Gynecology and Obstetrics, Aarhus University Hospital, Denmark. Consent was obtained from the mothers, but studies on anonymized samples, such as those used in the present study, are exempt from ethics permissions in Denmark (Kommitee-loven §§14. 3). CD34+ CB HSPCs (CB-HSPCs) were subsequently purified using EasySep Human Cord Blood CD34 Positive Selection kit II according to the manufacturer's instructions (STEMCELL Technologies). Briefly, a pre-enrichment of CD34+ cells was performed where bi-specific antibodies targeting unwanted cells were used during standard Ficoll-Hypaque (GE Healthcare) density-gradient centrifugation. CD34+ cells were subsequently isolated using anti-CD34 immunomagnetic beads (positive selection). CD34+ CB HSPCs were either freshly used or cryopreserved until use.

## Isolation of cHSPC from peripheral blood

Buffy coat samples were obtained from normal healthy donors from Aarhus University Hospital Blood Bank. CD34+ cHSPCs were subsequently purified using the EasySep Complete Kit for Human Whole Blood CD34+ Cells according to the manufacturer's instructions (STEMCELL Technologies). Briefly, a pre-enrichment of CD34+ cells was performed where bi-specific antibodies targeting unwanted cells were used during standard Ficoll-Hypaque (GE Healthcare) density-gradient centrifugation. CD34+ cells were subsequently isolated using anti-CD34 immunomagnetic beads (positive selection). CD34+ cHSPCs were cryopreserved until use.

## Pre-expansion of CB-HSPC or cHSPC

In order to pre-expand HSPCs, cells was seeded at LD ($1 \times 10^5$ cells/mL) in SFEM II (STEMCELL Technologies) or SCGM (CellGenix) to enable non-cGMP or cGMP conditions, respectively. Medium was supplemented with 20 µg/mL streptomycin and 20 U/mL penicillin (Gibco, Life Technologies), as well as 100 ng/mL of the growth factors Flt3-L, TPO, and SCF (PeproTech). In addition, medium was supplemented with the small molecule inhibitors StemRegenin 1 (1 µM, SR1, STEMCELL Technologies) and UM171 (35 nM, STEMCELL Technologies). Throughout the period of pre-expansion, cells were kept at LD ($1$–$5 \times 10^5$ cells/mL), and medium was replenished at least every third day. Cells were pre-expanded for up to 8 days before being cryopreserved using CryoStor10 (CS10, STEMCELL Technologies). Upon thawing, cells were validated for CD34 expression.

## Priming of HSPC-pDCs

Priming of HSPC-pDCs was performed as previously described (*Laustsen, 2018*). Isolated HSPC-pDCs were primed in the same medium as the differentiation was done in (either RPMI 1640, SFEM II, or DC medium). Medium was devoid of growth factors and only supplemented P/S and IL-3 (20 ng/mL). pDCs were primed with 250 U/mL IFN-β (PBL Assay Science) and 250 U/mL IFN-γ (PeproTech) or left unprimed. For HSPC-pDCs differentiated in DC medium, medium was also supplemented with 50 µg/mL AA. Cells were primed for 3 days before being phenotypically or functionally characterized.

## TLR7 or TLR9 agonist activation

To analyze the capacity of HSPC-pDCs to produce type I IFN, $4 \times 10^4$ pDCs were seeded out in 96-well plates in the same medium as the differentiation was done in (either RPMI 1640, SFEM II, or DC medium). Medium was devoid of growth factors and only supplemented with P/S and IL-3 (20 ng/mL). Cells were subsequently stimulated with agonists directed against TLR7 (R837, tlrl-imq, InvivoGen) or TLR9 (CpG-A 2216, tlrl-2216-1, InvivoGen) at a final concentration of 2.5 µg/mL. 20 hr post stimulation, supernatants were harvested and cryopreserved at –20 °C until analysis.

## Assessment of functional type I IFN

To quantify functional type I IFN, the reporter cell line HEK-blue IFN-α/β was utilized according to the manufacturer's instructions (InvivoGen). The cell line was maintained in DMEM + Glutamax-1 (Gibco, Life Technologies), supplemented with 10% heat-inactivated FCS, 100 µg/mL streptomycin and 200 U/mL penicillin (Gibco, Life Technologies), 100 µg/mL normocin (InvivoGen), 30 µg/mL blasticidin (InvivoGen), and 100 µg/mL zeocin (InvivoGen). Cells were passaged using 1× trypsin (Gibco, Life Technologies) and were not passaged more than 20 times. The HEK-blue IFN-α/β cell line has been

generated by stable transfection of HEK293 cells to express IRF9 and STAT2. Moreover, the cell line has been modified to express a reporter gene encoding secreted alkaline phosphatase (SEAP) under the control of ISG54 promoter. Activity of SEAP was assessed using QUANTI-Blue (InvivoGen). Color change was measured at an optical density (OD) of 620 nm using the SpectraMax iD3 platereader (Molecular Devices). For the generation of standard curve, hIFNa2 (PBL Assay Science) was used and ranged from 2 to 500 U/mL.

## Enzyme-linked immunosorbent assay

Protein levels of TNF-α, IL-6, and IL-12 (p70) were evaluated using the duoset ELISA kits from R&D Systems, following the manufacturer's instructions.

## Phenotypic analysis of cells using flow cytometry

Flow cytometry was used to immunophenotype pDCs. Briefly, cells were spun down and resuspended in 100 µL PBS. Cells were stained with Ghost Dye Red 780 Viability Dye (13-0865, Tonbo) for 30 min before being washed with FACS buffer (PBS with 2% FCS and 1 mM EDTA). Cells were subsequently resuspended in 50 µL FACS buffer and stained with the following antibodies: FITC anti-human Lineage Cocktail (CD3 [UCHT1], CD14 [HCD14], CD16 [3G8], CD19 [HIB19], CD20 [2H7], and CD56 [HCD56], 348801, BioLegend), APC anti-human CD11c (3.9, 20-0116 T100, Tonbo Biosciences), PE anti-human CD123 (6H6, 12-1239-42, eBioscience), PE-Cy7 anti-human CD303 (201a, 25-9818-42, eBioscience), BV605 anti-human CD2 (RPA-2.10, BioLegend), and BV650 anti-human HLA-DR (G46-6 BD Biosciences). Cells were stained for 30 min before being washed three times in FACS buffer. To validate the expression CD34$^+$ on HSPCs, cells were stained with PE-Cy7 anti-human CD34 (581, 343516, BioLegend) for 30 min, washed three times in FACS buffer before being resuspended in FACS buffer. PI was added to the cells as a viability dye in a concentration of 1:100 (13-6990 Tonbo Biosciences). Fluorescence intensities were measured using the Quanteon flow cytometer equipped with four lasers (405 nm, 488 nm, 561 nm, and 637 nm) and 29 photomultiplier (PMT) detectors or the NovoCyte flow cytometer with two lasers (488 nm and 561 nm) and 12 PMT detectors (ACEA Biosciences, Inc). Data analysis was done using FlowJo (version 10, Tree Star, Ashland, OR). Individual gating strategies are depicted in supplementary figures and outlined in figure legends.

## RNA-seq analysis

HSPC-pDCs were stored in RNAprotect Cell Reagent at –80° until total RNA was extracted using the RNeasy Plus Micro Kit (QIAGEN), which efficiently eliminates genomic DNA without the need for DNase treatment. The total RNA was sent to BGI Europe for RNA-seq. Here, a non-stranded and polyA-selected mRNA library was prepared from the total RNA and subjected to PE100 sequencing using the BGISEQ platform. The samples generated on average about 4.84 Gb bases per sample on DNBSEQ. Low-quality reads were filtered and the remaining reads were mapped to the genome with an average mapping ratio with the reference genome at 92.74%, the average gene mapping ratio at 79.90% . In total, 18,412 genes were identified. Gene expression was calculated based on the reads, and differentially expressed genes and gene ontology analyses were analyzed on BGI's software analysis platform Dr. Tom. Statistical significance was based on a Q-value below 0.05.

## Statistical analysis

All data were plotted using GraphPad Prism 8.0 (GraphPad Software, San Diego, CA). The data are shown as means of biological replicates ± standard error of mean (± SEM). Statistically significant differences between groups were determined using one-way or two-way ANOVA, followed by Bonferroni's post-hoc test. *$p \leq 0.05$, **$p \leq 0.01$, ***$p \leq 0.0001$.

## Acknowledgements

ROB gratefully acknowledges the support from a Lundbeck Foundation Fellowship (R238-2016-3349), the Independent Research Fund Denmark (9144-00001B), an AIAS-COFUND (Marie Curie) fellowship from the Aarhus Institute of Advanced Studies (AIAS) co-funded by Aarhus University's Research Foundation and the European Union's seventh Framework Program under grant agreement no 609033, the Novo Nordisk Foundation (NNF17OC0028894), Innovation Fund Denmark (8056-00010B), the Carlsberg Foundation (CF17-0129 and CF20-0424), Slagtermester Max Wørzner og Hustru Inger Wørzners

Mindelegat, the AP Møller Foundation, and the Riisfort Foundation. MRJ gratefully acknowledges the support from the Lundbeck Foundation Fellowship (R238-2016-2708), the Independent Research Fund Denmark (8020-00201B), and the Novo Nordisk Foundation (NNF18OC0053146). RMS received support from an AIAS-COFUND (Marie Curie) fellowship from the Aarhus Institute of Advanced Studies (AIAS) co-funded by Aarhus University's Research Foundation and the European Union's Horizon 2020 Research and Innovation Program under the Marie Sklodowska-Curie actions (grant agreement no. 754513). We would like to thank the FACSCore Facility at Aarhus University for providing invaluable help with flow cytometry. A special thanks to Ane Kjeldsen and Pernille Thornild Møller for technical assistance. Aarhus University has filed a patent related to this work with AL, MRJ, and ROB as co-inventors (patent application number 21183430.4). AL, MRJ, and ROB hold equity in the Danish company UNIKUM Therapeutics ApS. MRJ serves on the board of directors of UNIKUM Therapeutics ApS, and AL and ROB are part-time employees of UNIKUM Therapeutics ApS. ROB holds equity in Graphite Bio. *Figure 6* was made with a premium version of Biorender with a standard academic license to publish.

## Additional information

### Competing interests

Anders Laustsen: Aarhus University has filed a patent related to this work with AL, MRJ and ROB as co-inventors. AL holds equity in the Danish company UNIKUM Therapeutics ApS. AL is a full-time employee at UNIKUM Therapeutics ApS.. Martin R Jakobsen: Aarhus University has filed a patent related to this work with AL, MRJ and ROB as co-inventors. MRJ holds equity in the Danish company UNIKUM Therapeutics ApS. MRJ serves on the board of directors of UNIKUM Therapeutics ApS. Rasmus O Bak: Aarhus University has filed a patent related to this work with AL, MRJ and ROB as co-inventors. ROB holds equity in the Danish company UNIKUM Therapeutics ApS. ROB is a part-time employee at UNIKUM Therapeutics ApS. ROB holds equity in Graphite Bio.. The other authors declare that no competing interests exist.

### Funding

| Funder | Grant reference number | Author |
|---|---|---|
| Lundbeckfonden | R238-2016-3349 | Rasmus O Bak |
| Aarhus Institute of Advanced Studies, Aarhus Universitet | | Renée M van der Sluis Rasmus O Bak |
| European Union | 609033 | Rasmus O Bak |
| Lundbeckfonden | R238-2016-2708 | Martin R Jakobsen |
| Independent Research Fund Denmark | 8020-00201B | Martin R Jakobsen |
| Novo Nordisk Fonden | NNF18OC0053146 | Martin R Jakobsen |
| European Union | 754513 | Renée M van der Sluis |
| Carlsbergfondet | CF17-0129 | Rasmus O Bak |
| Carlsbergfondet | CF20-0424 | Rasmus O Bak |

The funders had no role in study design, data collection and interpretation, or the decision to submit the work for publication.

### Author contributions

Anders Laustsen, Conceptualization, Data curation, Formal analysis, Investigation, Methodology, Writing – original draft; Renée M van der Sluis, Albert Gris-Oliver, Sabina Sánchez Hernández, Ena Cemalovic, Data curation, Formal analysis, Writing – review and editing; Hai Q Tang, Lars Henning Pedersen, Niels Uldbjerg, Resources, Writing – review and editing; Martin R Jakobsen, Rasmus O Bak, Conceptualization, Formal analysis, Funding acquisition, Methodology, Project administration, Resources, Supervision, Writing – review and editing

## Author ORCIDs
Anders Laustsen http://orcid.org/0000-0002-0306-1736
Renée M van der Sluis http://orcid.org/0000-0002-7668-2517
Albert Gris-Oliver http://orcid.org/0000-0003-1802-9541
Niels Uldbjerg http://orcid.org/0000-0002-6449-6426
Martin R Jakobsen http://orcid.org/0000-0001-8847-9201
Rasmus O Bak http://orcid.org/0000-0002-7383-0297

## Ethics

Human subjects: De-identified umbilical cord blood (UCB) samples were obtained following scheduled caesarean section deliveries of healthy infants at Department of Gynecology and Obstetrics, Skejby University Hospital. Consent was obtained from the mothers, but studies on anonymized samples, such as those used in the present study, are exempt from ethical permissions in Denmark (Kommitee-loven § §14. 3). Buffy coat samples were obtained from normal healthy donors from Aarhus University Hospital Blood Bank. These were de-identified samples and studies on anonymized samples are exempt from ethical permissions in Denmark (Kommiteeloven § §14. 3).

## Decision letter and Author response
Decision letter https://doi.org/10.7554/eLife.65528.sa1
Author response https://doi.org/10.7554/eLife.65528.sa2

## Data availability

All data generated or analysed during this study are included in the manuscript and supporting files. Sequencing data have been deposited in Dryad (doi: https://doi.org/10.5061/dryad.69p8cz92zz).

The following dataset was generated:

| Author(s) | Year | Dataset title | Dataset URL | Database and Identifier |
|---|---|---|---|---|
| Laustsen A | 2021 | Data From: Ascorbic acid supports ex vivo generation of plasmacytoid dendritic cells from circulating hematopoietic stem cells | http://dx.doi.org/10.5061/dryad.69p8cz92z | Dryad Digital Repository, 10.5061/dryad.69p8cz92z |

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
