## [Decision Letter]

**Acceptance summary:**

This manuscript describes a robust method for generating GMP HSPC‐pDCs from HSPCs obtained from standard blood samples without the need for mobilization regiments like G‐CSF and plerixafor. The group have found that supplementation of ascorbic acid (AA) to the cGMP medium was key for achieving high and consistent numbers of highly functional HSPC‐pDCs. Tha authors also include downstream effector function analysis, The study will be important for the design of future experiments and thus very impactful.

**Decision letter after peer review:**

Thank you for submitting your article "Ascorbic acid supports ex vivo generation of plasmacytoid dendritic cells from circulating hematopoietic stem cells" for consideration by *eLife*. Your article has been reviewed by 2 peer reviewers, and the evaluation has been overseen by Carlos Isales as the Senior and Reviewing Editor. The reviewers have opted to remain anonymous.

The work has potential but requires essential additional data especially concerning how ascorbic acid affects type 1 IFN production in pDCs and whether it contributes to metabolic fitness of pDCs via ROS removal. As it stands the manuscript lacks crucial explanation for the AA's mechanism of action in pDC cultures.

In this manuscript Laustsen and colleagues describe an optimized protocol to obtain pDCs for therapeutic purposes. A common problem of cell based therapies is setting the ex vivo conditions optimally to provide the patient-derived cells at high enough numbers for performing adoptive transfers. This is further complicated by prolonged cultures that may lead to reduced functionality of the cells in other words "exhaustion". The authors nicely build on their previous work to address these common problems for pDC based therapies by carefully adjusting the ex vivo differentiation conditions of HSPCs. They successfully demonstrate that the combination of a low density culture condition, maintained with more frequent cell splitting, and an engineered GMP compliant procedure via addition of ascorbic acid to media elicits consistently high numbers of highly functional pDCs. Strikingly, this method is applicable to HSPCs derived by simple aphaeresis from peripheral blood, which increases the practicality of the method. The authors provide evidence for the reduced functionality of pDCs in standart cGMP media and they managed to solve this problem by adding ascorbic acid into the culture medium. While these findings are extremely relevant to clinical application of pDC-based immunotherapy, one wonders the mechanistic explanation for why and how ascorbic acid exerts such an isolated effect on pDC functionality.

*Reviewer #1:*

The manuscript by Laustsen et al., describes a method for a robust generation GMP HSPC‐pDCs. The data is highly significant and the study is done in a rigorous way, experiments are validated in several replicates and the manuscript is well written. in addition, the addition of ascorbic acid for the recovery of IFNa response is fascinating. The major weakness is that no downstream analysis was done to understand the impact of the AA, or the effector function of the cells. The authors' claims and conclusions are justified by their data.

A couple of point added will strengths the manuscript:

1. Class-II expression on differentiated HSPC‐pDCs (Lin‐CD11c‐CD123+CD303+) will be shown.

2. Performing rnaseq to determine a more global similarities and differences between the gmp-pdc +/- AA to try and evaluate the mechanism underlying the AA ability to restore IFN production in theses cells.

3. Alternatively, evaluate the expression levels of TLRs affected by AA.

4. What other cytokines are being produced by HSPC‐pDCs, IL-12?

5. Optional: do GMP HSPC‐pDCs express CD2 and do they possess any effector function

*Reviewer #2:*

Here in this manuscript, authors describe a robust culturing method for pDC differentiation that would be highly beneficial to a large audience in the cancer immunology field. However, this study would be more appropriate to publish a methodology manuscript than a research manuscript as it lacks crucial information on the effects of ascorbic acid on pDCs. Therefore, this study would highly benefit from further elucidating the role of ascorbic acid in pDCs especially via exploring its possible effects on Type 1 IFN production and alleviating the potential harmful effects of ROS. Albeit useful, the presented data don't warrant an impactful publication in *eLife*.

---

## [Author Response]

Reviewer #1:The manuscript by Laustsen et al., describes a method for a robust generation GMP HSPC‐pDCs. The data is highly significant and the study is done in a rigorous way, experiments are validated in several replicates and the manuscript is well written. in addition, the addition of ascorbic acid for the recovery of IFNa response is fascinating. The major weakness is that no downstream analysis was done to understand the impact of the AA, or the effector function of the cells. The authors' claims and conclusions are justified by their data.

We are glad to hear the reviewer finds the method for generating HSPC-pDCs highly significant. We believe the new experimental additions have addressed the reviewer’s comments and we hope the study is now appropriate for publication.

A couple of point added will strengths the manuscript:1. Class-II expression on differentiated HSPC‐pDCs (Lin‐CD11c‐CD123+CD303+) will be shown.

We thank the reviewer for the suggestion. We have now added analyzes of the expression of HLA-DR on the HSPC-pDCs (Figure 4—figure supplement 1r-s), and added RNAseq data on expression levels of genes related to antigen processing and presentation, including HLA genes (Figure 5c and Figure 5—figure supplement 1a).

2. Performing rnaseq to determine a more global similarities and differences between the gmp-pdc +/- AA to try and evaluate the mechanism underlying the AA ability to restore IFN production in theses cells.

This is a highly relevant point. Accordingly, we have performed an RNAseq analysis where we have found that AA addition changes the expression of 1250 genes (Figure 5). Interestingly, we find that several genes related to pDC development and function are highly upregulated by AA addition. Of these, many are associated with TLR7 and TLR9 processing, trafficking, and innate immune pathways, including IRF5, IRF7, IRF8, TLR7, TLR9, MyD88, TRAF3, UNC93B1, LAMP5, PACSIN-1, RSAD2, SLC15A4 and SPP1 (Figure 5c), providing a plausible explanation for the restorative function AA has on the IFN production in the generated HSPC-pDCs.

3. Alternatively, evaluate the expression levels of TLRs affected by AA.

Addressed above in point 2.

4. What other cytokines are being produced by HSPC‐pDCs, IL-12?

We thank the reviewer for raising this point. We have now investigated the production of TNF-α, IL-6, and IL-12 by HSPC-pDCs upon TLR7 and TLR9 stimulation (Figure 4—figure supplement 2l-m and t). Interestingly, we find that TNF-α and IL-6 production is also affected negatively if AA is not present, providing further evidence that AA exerts a general effect on the TLR7 and TLR9 pathways.

For IL-12, no detectable levels were observed as expected since pDCs are known to be incapable of producing IL-12 in response to TLR agonists. A study previously determined that IL-12 production in purified pDCs is derived from contaminating precursor conventional dendritic cells (pre-DCs) (See et al., Science, 2017). The same study showed that pre-DCs express many markers that overlap with pDCs, including CD123, CD303 and CD304, but uniquely express CD2, CD5, and CD33. Related to question 5 below, we therefore also investigated surface levels of CD2 on isolated HSPC-pDCs and confirmed that no pre-DCs are present within the immunomagnetically-selected HSPC-pDC population (Figure 4—figure supplement 2u).

5. Optional: do GMP HSPC‐pDCs express CD2 and do they possess any effector function

Addressed above in point 4.

Reviewer #2:Here in this manuscript, authors describe a robust culturing method for pDC differentiation that would be highly beneficial to a large audience in the cancer immunology field. However, this study would be more appropriate to publish a methodology manuscript than a research manuscript as it lacks crucial information on the effects of ascorbic acid on pDCs. Therefore, this study would highly benefit from further elucidating the role of ascorbic acid in pDCs especially via exploring its possible effects on Type 1 IFN production and alleviating the potential harmful effects of ROS. Albeit useful, the presented data don't warrant an impactful publication in eLife.

We highly appreciate that the reviewer finds the study to be of high benefit to the large community within the field of cancer immunology and thank the reviewer for the suggestions. We agree that the study at the time of the initial submission was focused on the methodology and would have benefitted from additional data supporting the role of ascorbic acid in HSPC-pDCs. Consequently, we have now performed RNAseq on HSPC-pDCs generated with and without ascorbic acid. We find that differentially expressed genes are involved in pDC development and function (Figure 5 and Figure 5—figure supplement 1), including several related to TLR7 and TLR9 innate immune responses, explaining the impact of AA on the TLR7 and TLR9 innate immune responses. In contrary, we did not find any differentially expressed genes related to ROS and oxidative stress. With the new experimental additions to the study, we believe we have addressed the reviewer’s comments and hope the study is appropriate for publication.